# The Effects of Vitamin D on the Breast Cancer Tumor Microenvironment

**DOI:** 10.3390/cancers17233751

**Published:** 2025-11-24

**Authors:** Balquees Kanwal, Syeda Saba Shah, Farzana Shaheen, Mekonnen Sisay Shiferaw, Deepanshu Maurya, Yujie Li, Saranya Pounraj, Zaklina Kovacevic

**Affiliations:** 1Atta-ur-Rahman School of Applied Biosciences (ASAB), National University of Sciences and Technology (NUST), H-12, Islamabad 44000, Pakistan; bkanwal.phdabs19asab@student.nust.edu.pk (B.K.); sshah.phdabs18asab@student.nust.edu.pk (S.S.S.); 2Department of Chemistry, Faculty of Science, Allama Iqbal Open University (AIOU), H-8/4 Campus, Islamabad 44310, Pakistan; farzana.shaheen@aiou.edu.pk; 3School of Biomedical Sciences, University of New South Wales, Sydney 2052, Australia; m.shiferaw@unsw.edu.au (M.S.S.); d.maurya@student.unsw.edu.au (D.M.); yujie.li7@unsw.edu.au (Y.L.); s.pounraj@unsw.edu.au (S.P.)

**Keywords:** breast cancer, tumor microenvironment, vitamin D, fibroblasts

## Abstract

Breast cancer (BC) develops within a complex tumor microenvironment (TME) consisting of cancer-associated fibroblasts, adipocytes, immune cells, cancer stem cells, and the extracellular matrix, all of which play a crucial role in tumor progression and metastasis. Vitamin D (VD) is widely recognized for its protective effects against BC, due to its ability to regulate the cell cycle, promote apoptosis, inhibit angiogenesis, and suppress metastasis. While the anti-cancer properties of VD have been extensively studied, its specific role within the BC TME remains underexplored. This review highlights VD’s regulatory influence on various TME components, underscoring its potential to reshape the TME and enhance BC treatment strategies.

## 1. Introduction

Breast cancer is the second most frequent cause of mortality in women globally, with 2.3 million BC cases diagnosed and 670,000 deaths in 2022 [1]. With steadily rising diagnoses rates each year, this staggering upsurge in BC poses a serious health challenge for healthcare authorities. The complex and heterogenous nature of BC remains a major hurdle in terms of its treatment [2,3]. Currently, the treatment strategies used for BC include chemotherapy, surgical resection, targeted hormonal and radiation therapy, although development of resistance and metastatic progression lead to high rates of mortality [4]. In fact, of the 63% of patients diagnosed with localized BC, a third will eventually progress to metastatic cancer which rapidly becomes refractory to current therapies [5].

The TME plays a crucial role in the development and progression of BC. Consisting of various cell populations including stromal cells, connective tissue, tumor vasculature, immune cells and the extracellular matrix (ECM) [4,6], the TME is continuously evolving as the tumor progresses. Tumor cells secrete various factors, including cytokines, chemokines and extracellular vesicles, which can influence the TME. In response, TME stromal cells engage in metabolic cross talk with BC cells, activating angiogenesis, cell adhesion, migration, invasion and metastasis [7,8]. Biophysical characteristics of the TME, like stiffness of the ECM and a hypoxic environment, can directly influence metastatic invasion of cancer cells [9]. Recognizing the importance of the TME in BC progression, there has been an exponential increase in the development of novel treatment approaches that target different elements of the TME, including immune checkpoint inhibitors and monoclonal antibodies [10]. VD is a relatively underexplored factor in the BC TME, although emerging evidence suggests it may hold promise as a therapeutic agent by modulating the TME to influence disease outcomes.

It has been well-established that VD deficiency is linked to a significantly increased risk of developing BC, as well as being inversely correlated with poor prognosis and metastatic progression in BC patients [11]. VD exerts its anti-tumor effects through multiple signaling pathways, encompassing the inhibition of cell proliferation, promotion of differentiation, induction of apoptosis, and suppression of both angiogenesis and metastasis [12,13]. However, recent studies have demonstrated that VD does not solely impact tumor cells; it profoundly influences diverse components within the TME. Given its substantial anti-cancer influence on both cancer cells and the TME, VD is emerging as a potential therapeutic agent capable of re-programming the TME to aid in clinical treatment and prognosis. [14]. This review aims to give a comprehensive and up-to-date summary of how VD influences the different TME elements in BC, and its potential utility in enhancing current therapeutic approaches by re-shaping the BC TME.

## 2. Methodology

A comprehensive electronic literature search was conducted on legitimate databases including PubMed, Scopus, Web of Science, MEDLINE and Google Scholar using the keywords “Vitamin D”, “tumor microenvironment”, “breast cancer”, “vitamin D metabolism”, “vitamin D receptor”, “vitamin D signaling”, “breast cancer progression”, and “therapeutic role of vitamin D”. Both original research articles and review papers were included. The search covered research papers until 2025 in English. In addition, data from ClinicalTrials.gov and related clinical trial registries were also visited to include completed and ongoing studies evaluating the effects of VD in BC. Studies utilizing experimental models like BC cell lines, animal models, patient derived ex vivo samples, and 3D culture systems were included. Non-English, irrelevant or male BC studies were excluded. Reference lists of the retrieved articles were also screened to identify additional relevant studies.

## 3. VD Biogenesis and Metabolism

Prior to discussing the role of VD in BC, it is important to give a brief introduction to VD biogenesis and metabolism. Multiple forms of VD have been reported, including: ergocalciferol (vitamin D_2_), cholecalciferol (Vitamin D_3_), 22-dihydroergocalciferol (Vitamin D_4_) and sitocalciferol (Vitamin D_5_). Another source of VD is ergosterol, a provitamin form of VD that is primarily found in fungal cell membranes, and which can be converted into ergocalciferol upon UV exposure [15]. Cholecalciferol (vitamin D_3_) and ergocalciferol (vitamin D_2_) are the most biologically and biochemically significant forms of VD [16]. Despite their chemical similarity, their biological origins differ. Vitamin D_3_ is synthesized endogenously in the skin upon exposure to UVB radiation and accounts for approximately 95% of the body’s vitamin D requirements, whereas vitamin D_2_ is obtained from dietary sources [17].

The biologically active form of VD, 1α,25-dihydroxyvitamin D_3_ (calcitriol), is produced through two hydroxylation steps: first in the liver and then in the kidney. The initial hydroxylation occurs in the liver via CYP2R1 and CYP27A1, converting cholecalciferol to 25-hydroxyvitamin D_3_ (calcidiol). The second hydroxylation takes place in the kidney, where CYP27B1 converts calcidiol to 1α,25-dihydroxyvitamin D_3_ (calcitriol) [3]. To regulate circulating levels of calcitriol, the enzyme CYP24A1 hydroxylates both calcitriol and calcidiol at carbon 24, facilitating their excretion via urine or feces [18]. Although over 50 VD metabolites have been identified, calcitriol is the principal form recognized as biologically active in systemic circulation [19].

###  VD Role in Normal Breast Homeostasis

To exert its effects, VD binds to the vitamin D receptor (VDR), followed by the formation of a heterodimer with the retinoic acid receptor [20]. This complex can bind to the VD response element (VDRE) in the promoter region of target genes, leading to effects such as apoptosis, cell proliferation, autophagy and differentiation [21]. Additionally, studies in mice have shown that VDR-mediated signaling plays a significant role in regulating mammary cell turnover through the reproductive cycle. VDR-null mice exhibit accelerated mammary gland development during pregnancy and delayed involution post-lactation, indicating that VDR contributes to both differentiation and apoptosis in mammary tissue [22]. It is noteworthy that in breast tissue, VD signaling can be regulated by hormones such as estrogen, phytoestrogens and retinoids, which up-regulate VDR gene expression [23]. The presence of the enzyme 1α-hydroxylase (CYP27B1) in normal breast tissue suggests a potential mechanism by which VD may exert protective effects against cancer [24,25]. Calcitriol is locally synthesized in breast epithelial cells from calcidiol viathe action of CYP27B1. This locally produced calcitriol does not enter systemic circulation but instead acts within the same tissue where it is synthesized, supporting autocrine or paracrine signaling [25].

## 4. Vitamin D and Breast Cancer

The association between BC susceptibility and VD intake has been well studied, with an inverse relationship found between VD intake and BC risk [26]. In a cross-sectional study of Brazilian post-menopausal BC patients, low serum VD levels were significantly correlated with poorer clinical outcomes [27]. Moreover, VD supplementation (>400 IU/day) was found to be associated with reduced risk of BC [28]. Another study demonstrated that deficiency of VD increased BC susceptibility by 23% in African American women [29]. Various factors like obesity, age, inadequate physical activity, skin type, living at high altitude, race and smoking can cause reduced levels of VD [30]. In fact, multiple studies examining VD levels of newly diagnosed BC patients report that 65–95% were VD deficient [31,32,33], with one study reporting that only 7% of the 105 women assessed had sufficient VD levels [31]. Notably, another study demonstrated that BC patients diagnosed with the more aggressive triple negative BC (TNBC) subtype had significantly lower serum VD levels than those with estrogen receptor positive (ER^+^) BC, with both cancer groups having lower VD levels than those with benign breast disease [34]. This reflects a widespread and largely unrecognized worldwide epidemic, with VD deficiency being prevalent across many different cancer types [35].

In addition to it classical role in regulating calcium homeostasis [11], VD also promotes cell differentiation, inhibits proliferation of specific cells and prevents angiogenesis [36], which may account for its reported anti-cancer properties. Further, VD can induce apoptosis [37], and has anti-metastatic and anti-inflammatory effects [38] (Figure 1).

In mammary cells, VD and its receptor VDR regulate the optimal development of mammary glands and impede breast tumor growth in animal models [39]. Upon ligand binding, the activation followed by nuclear transfer of VDR influences BC cell biology by blocking epithelial–mesenchymal transformation (EMT), infiltration, metastasis and autophagy [39]. The higher expression of VDR in the nuclei of BC cells is linked with decreased risk of death due to BC and is associated with favorable prognostic factors [40]. However, cellular factors triggered during tumorigenesis can alter the down-stream function of VDR. For instance, Ras like oncogenes and transcription factors promoting EMT interfere with VDR signaling, eliciting desensitization of VD action [41]. Further, oncogenic signaling pathways including ERBB2/ERK/AKT, WNT/β-Catenin, JAK/STAT, ERα and NF-κB can reduce VDR levels, thereby reducing the tumor suppressive effect of VD [42]. These findings endorse the idea of VD signaling being an important regulator of physiological development of mammary glands, the deregulation of which leads to alterations progressing towards carcinogenesis [42].

### 4.1. VD Role in Cancer Cell Proliferation

Numerous studies have shown that compounds of VD have a significant role in altering the cell cycle in various cell systems [43]. Using MCF-7 BC cells, which express low levels of VDR, Veeresh et al. demonstrated that VD treatment significantly increased the sub-G0/G1 cell population compared to vehicle-treated controls. This was accompanied by a reduction in the S and G2/M phase populations, indicating a dose-dependent inhibition of cell growth by VD [44,45]. Additionally, VD significantly reduced the expression levels of the proliferation marker Ki-67 in MCF-7 and MDA-MB-231 cells [46]. The anti-proliferative effects of VD were exploited in a recent study examining ER^+^ BC, where the VD analog inecalcitol was combined with CDK4/6 inhibitor palbociclib and found to significantly reduce tumor growth in vivo, being more potent than palbociclib alone [47]. Notably, this latter study also found that the same treatment combination did not improve efficacy of palbociclib in triple negative BC when examined in vivo.

The growth inhibitory effects of VD have been further demonstrated using the transgenic MMTV-PyMT (mouse mammary tumor virus promoter-driven polyoma middle T oncoprotein) mouse model, which recaptures human BC progression in vivo [48]. A low VD diet led to significant acceleration of mammary neoplasia in the MMTV-PymT mice, while systemic perfusion of VD delayed tumor appearance and decreased lung metastases. This was correlated with significantly reduced Ki-67, cyclin D1 and ErbB2 levels in tumors [48].

The underlying mechanisms by which VD inhibits BC proliferation could be linked to its ability to induce the expression of p21^WAF1/CIP1^ and p27^kip1^, which mediate cell cycle arrest [49]. Moreover, VD can stimulate the P15 and P16 like inhibitors of cyclin dependent kinases [50], while inhibiting the growth-regulating RAS-RAF-MEK-ERK pathway [46]. However, others have shown that VD induces apoptosis in BC cells and suggest this as the main driver behind its anti-cancer effects in vivo [38].

### 4.2. VD Role in Apoptosis

A plethora of studies have demonstrated a role for VD in apoptosis due to its effects on the expression of numerous pro- and anti-apoptotic proteins [51]. Veeresh et al. demonstrated that VD induced apoptosis via the induction of caspase-3/7 in MCF-7 and MDA-MB-231 cells [44]. In another study, treatment of MCF-7 and TNBC cells with VD analog inecalcitol significantly increased apoptosis in MCF-7 cells, while no effect was observed in TNBC cells [47]. The ability of VD to induce apoptosis is mediated by its inhibitory effects on anti-apoptotic factors such as Bcl-2 and Bcl-XL, while upregulating pro-apoptotic counterparts, namely Bax and Bak [3]. Additionally, VD also targets multiple pathways that have anti-apoptotic effects, including the RAS/MEK/ERK signaling pathway in MCF-7 and MDA-MB-231 cells [18]. Additionally, the VDR is a direct transcriptional target of p53 and its family members (e.g., p63, p73). Wild-type p53 (wtp53) enhances VDR expression and promotes VD-mediated pro-apoptotic responses. Conversely, mutant p53 (mutp53) can physically interact with VDR, altering its transcriptional activity and converting VD signaling into a pro-survival pathway in cancer cells. This interaction may contribute to VD resistance in tumors harboring p53 mutations [52].

### 4.3. VD Role in Inhibition of Angiogenesis

Another significant anti-cancer function of VD is its ability to inhibit angiogenesis. Calcitriol was found to attenuate endothelial cell sprouting and proliferation by inhibiting vascular endothelial growth factor (VEGF) expression in vitro. This led to significantly reduced tumor vascularization in a BC xenograft mouse model where ER^+^ MCF-7 cells over-expressing VEGF were treated with calcitriol over 8 weeks [53]. This was further confirmed in a TNBC xenograft model, where tumor onset, volume and microvascular density were all reduced in mice administered calcitriol combined with curcumin [54]. This same study also showed that a combination of calcitriol and resveratrol also reduced the number of microvessels in vivo. The co-administration of curcumin or resveratrol was hypothesized to synergistically enhance the anti-angiogenic effects of calcitriol, as either compound alone did not exert significant anti-angiogenic effects [54].

### 4.4. Anti-Metastatic and Anti-Invasive Role of VD

Several studies have demonstrated the ability of VD to inhibit BC cell invasion and metastasis [55,56]. VD reduced the release of prostaglandin in BC [57], which plays an important role in triggering cancer cell proliferation and invasion [58], while also mediating resistance to apoptosis. This effect was mediated by the reduced expression of cyclooxygenase 2 (COX-2; [57]), which is responsible for the prostaglandin release [21]. VD was also involved in catalyzing prostaglandin conversion into inactive ketone compounds by escalating the expression of enzyme 15-hydroxy prostaglandins dehydrogenase [21]. Liu et al. showed that using a calcitriol nanosystem to deliver high doses of calcitriol to BC cells in mice significantly reduced tumor growth and had strong anti-metastatic effects [59]. VD was also found to suppress metastasis in BC cells by up-regulating E-cadherin expression and down-regulating mesenchymal markers P-cadherin and N-cadherin [60].

### 4.5. VD Role in Autophagy

The VDR was found to act as a master regulator of autophagy at the transcriptional level, with activation by VD inducing an autophagic transcriptional signature in BC cells and being linked to better patient prognosis [61]. In addition, treatment with VD triggered autophagy, followed by autophagy-dependent cell death in MCF-7 BC cells, an effect found to be mediated by increased cytosolic free Ca^2+^ levels [62]. Specifically, the VD-mediated autophagy induction involved AMP-activated protein kinase (AMPK) activation, which was stimulated by Calcium/Calmodulin-Dependent Protein Kinase Kinase (CAMKK2) in MCF-7 cells [62]. Notably, the VDR is known to constitutively repress autophagy in BC, but upon VD stimulation, an increase in the autophagy basal levels occurs by the de-repression of MAP1LC3B (LC3B), a key autophagy gene [61]. In addition to its effects in BC, VD is also capable of inducing autophagy in neurons [63,64]. Table 1 presents a summary of the anti-cancer effects of VD, highlighting its role in various pathways and its regulation of specific proteins and genes.

## 5. Role of VD in the Tumor Microenvironment

The TME comprises a diverse array of stromal cells, including cancer-associated fibroblasts (CAFs), adipocytes, endothelial cells, immune cells, along with extracellular components such as the extracellular matrix and cytokines. VD is capable of modulating the stromal cells in the TME leading to suppression of tumor angiogenesis and metastasis [14]. Notably, the TME significantly influences therapeutic efficacy and has emerged as a pivotal target for the treatment of tumors. Given the anti-cancer properties of VD that have been highlighted above, in this section we delve into the role of VD within the BC TME, recognizing its potential significance in influencing cancer treatments and prognostic outcomes.

### 5.1. Cancer Associated Fibroblasts

The most prominent element within the BC TME are the CAFs [76]. Their involvement has been strongly linked to tumor malignancy and the facilitation of cancer progression [77,78,79]. CAFs originate from precursor cells like normal fibroblast (NFs), endothelial or epithelial cells, mesenchymal stem cells, adipocytes, vascular smooth muscle cells, pericytes and bone marrow fibrocytes [80,81,82,83]. The reciprocal crosstalk between cancer cells and fibroblasts leads to diverse biological and morphological alterations in CAFs throughout the course of tumor progression [84,85] (Figure 2). These transitions are integral to sustaining an optimal microenvironment that supports the survival and advancement of the tumor [86]. Research inquiries into understanding the function of CAFs have indicated that exclusively targeting tumor cells therapeutically is inadequate for comprehensive cancer treatment [87]. Particularly, BC CAFs can constitute up to 80% of the tumor mass and are key players in initiation and progression of BC [88,89]. Consequently, effective cancer therapy should involve concurrent targeting of the tumor microenvironment, particularly focusing on CAFs as a molecular target.

Although research indicates that CAFs have a role in promoting tumor development, there is also evidence that CAFs have a tumor suppressive effect [90]. The dual action of CAFs on tumor development can be linked to the diverse CAF characteristics [91]. The classification of neoplastic fibroblasts is often based on the expression of markers and is related to the cancer subtypes from which they are isolated [92]. Alpha-smooth muscle actinin (α-SMA), a smooth muscle cell marker expressed by actively proliferating cells like myofibroblasts, enables them to enhance the synthesis of ECM components, promote active contraction of wound edges and remodeling to support healing [92]. Furthermore, it was observed that α-SMA^High^ CAFs indirectly affect the immune response through ECM deposition, and cause matrix reconstruction where immune cells migrate or localize [93].

Recent studies have identified that multiple, functionally distinct CAF-subtypes exist in BC, distinguished by their expression of various markers. For instance, Brechbuhl et al. demonstrated that ER^+^ BC consists of two CAF subtypes distinguished by their expression of CD146 (also known as melanoma cell adhesion molecule (MCAM)) [94]. This latter study showed that CD146^−^ CAFs caused lower ER expression in BC cells, reducing estrogen sensitivity and increasing resistance to tamoxifen. Notably, BC patients with predominantly high levels of CD146^−^ CAFs were found to have poorer outcomes [94]. In contrast, CD146^+^ CAFs sustained tamoxifen sensitivity and were associated with fewer metastases [95]. Another study reported that CD10^+^ GPR77^+^ CAFs are associated with chemoresistance and stemness of BC through sustained secretion of NF-κB-dependent IL-6 and IL-8 [96]. Further, Costa et al. characterized four subsets of CAFs in BC with different properties and activation levels, designating them into CAF-S1, CAF-S2, CAF-S3 and CAF-S4 sub-types based on their expression of 6 markers including CD29, FAP, α-SMA, PDGFRβ, S100-A4/FSP1 and caveolin-1 [91]. They showed that CAF-S1 and CAF-S4 subsets accumulated differently in TNBC. Intriguingly, CAF-S1 promoted an immunosuppressive microenvironment and increased the survival of regulatory T cells, leading to inhibition of effector T cells. The CAF-S2 subtype was enriched in luminal A type BC, while the CAF-S4 subtype was enriched in HER2+ BC [91]. Finally, a recent study by Cords et al. analyzed the scRNAseq data from 14 BC patients and identified 9 CAF populations, with the most common sub-types being matrix CAFs (mCAFs), inflammatory CAFs (iCAFs), vascular CAFs (vCAFs), tumour-like CAFs (tCAFs), interferon response CAFs (ifnCAFs) and antigen presenting CAFs (apCAFs) [97].

Although CAFs are now widely recognized to significantly influence the whole process of BC carcinogenesis and response to anti-cancer therapies, research into the effect of potential treatments such as VD on CAFs is scarce. To address this gap, we have actively sought out studies that investigate how VD may influence CAFs and their different phenotypes in BC. The first evidence that VD may influence CAFs comes from a 1995 study that co-cultured BC fibroblasts with either cancer cells (MCF-7, BT-20) or non-cancerous cells (NPM-21T). While CAFs enhanced the proliferation of the cancer cells, the addition of VD metabolite calcitriol inhibited this effect [98]. These early observations were further confirmed in 2013, when the effects of calcitriol on both CAFs and matched normal fibroblasts (NFs) extracted from 5 BC patient tumors were examined [99]. This study demonstrated that VD treatment at supra-physiological levels of 100 nM for 24 h induced extensive transcriptional effects in BC CAFs and NFs, both of which express the VDR, although CAFs responded more robustly to VD. In CAFs, multiple proliferation genes including Neuregulin 1 (NRG1), Wingless-type family member 5A (WNT5A) and Platelet-derived growth factor C (PDGFC) were down-regulated, while Superoxide dismutase 2 (SOD2)*,* which protects cells from redox stress, was potently up-regulated. In NFs, VD increased expression of genes involved in cell proliferation and anti-apoptosis. Common genes that were up-regulated by VD in both CAFs and NFs included those involved in immune response and inflammation. This was the first evidence that VD might be reprogramming fibroblast behavior in BC.

This same team later used a more physiological model of BC to validate the transcriptional effects of VD, employing fresh BC tumor slices that were exposed to physiological (0.5 nM) concentrations of calcitriol for 24 h [100]. Under these conditions, the effects of VD on both cancer cells and CAFs were less intense, although a number of genes including CYP24A1, dipeptidyl peptidase-4 (DPP4) and carbonic anhydrase 2 (CA2) were still markedly up-regulated by VD in CAFs, suggesting that even physiological levels of VD given over a relatively short time-period are able to modulate the stroma in BC [100].

More recently, the effects of VD on BC CAF phenotype, protein expression and secretion were examined in the context of different clinical characteristics, including pre- and post-menopausal patients, VD-deficient and non-deficient patients, and those with or without metastatic disease [101]. CAFs isolated from 91 patients were treated with VD metabolite calcitriol ex vivo, which led to significantly reduced cell viability at doses ranging from 10 to 1000 nM. However, only doses ≥100 nM of calcitriol were able to significantly reduce viability of CAFs isolated from pre-menopausal women or those deficient in VD. Further, the CAF-mediated secretion of CCL2, a cytokine that recruits immunosuppressive cells into the TME, was also potently reduced by calcitriol treatment in all clinical groups. Overall, calcitriol was found to decrease the immunosuppressive phenotype of CAFs derived from BC patients. However, some differences were observed between the various clinical groups, with CAFs isolated from metastatic tumors responding more favorably to calcitriol when compared to their non-metastatic counterparts. This study also found that calcitriol reduced the pro-metastatic effects of CAFs on MCF-7 and MDA-MB-231 BC cells. These suggest that VD has potentially important effects on CAF phenotype and may influence their interaction with BC cells [101].

It is important to note that although VD generally promoted anti-cancer effects in CAFs, the effects of VD on BC CAFs were not clearcut, with the aforementioned studies also describing some pro-cancer effects which may be mediated by CAF heterogeneity, clinical characteristics and the patients’ VD status. For instance, CAFs derived from patients deficient in VD responded to calcitriol by increasing CXCL12, an immunosuppressive cytokine that promotes BC metastasis [101]. Another study examining the effects of VD supplementation or deficiency in murine models of BC revealed even further complexity [102]. In this latter study, NFs and CAFs from mice bearing three different mammary gland tumors (4T1, 67NR, and E0771) were assessed following either a VD normal diet (1000 IU), VD-deficient diet (100 IU) or VD-supplemented diet (5000 IU). In the tumor bearing mice, the effect of various VD treatments on NFs was dependent upon the tumor implanted and the mouse strain studied. For 67NR and E0771, no significant effect was found on NFs. But in highly metastatic 4T1 tumor bearing mice, fibroblast activation was observed in all mouse groups with calcitriol administration, with this effect being more pronounced in mice which were on a VD deficient diet. This latter group of mice had increased activation of NFs into CAFs. Although this study demonstrated variable results depending on the tumor type and mouse strain used, the results do suggest that maintaining sufficient VD levels is important, as additional supplementation can have undesirable effects in the context of BC if there is an underlying VD deficiency [102]. Thus, the effect of VD supplementation on CAFs in BC appears to be dependent on multiple factors including the underlying VD plasma levels and disease stage.

While studies investigating the effects of VD on BC CAFs are limited to the ones discussed above, multiple studies have demonstrated significant effects of VD on pancreatic cancer (PaC) CAFs. Pancreatic stellate cells (PSCs) which can convert into CAFs upon activation [103], were found to have high expression levels of VDR [103]. In fact, VDR was found to act as a master regulator of PSC activation, with VDR induction by its ligand VD analog calcipotriol promoting the quiescent state in PSCs in vitro and attenuating inflammation and fibrosis in vivo [103]. At the molecular level, calcipotriol treatment of PSCs reduced the expression of a host of genes with functional significance in the TME including inflammatory cytokines and growth factors [103]. Calcipotriol was also able to inhibit cross-talk between PSCs and cancer cells, and improved the response of PC tumor bearing mice to gemcitabine [103]. These results were further validated by others, clearly demonstrating that VD can inhibit the cross-talk between PC cells and CAFs, while also re-programming CAFs into a less tumor-promoting phenotype and reducing CAF-mediated immunosuppression [104,105]. The anti-cancer effects of VD were further validated in colorectal cancer (CRC), where VDR expression in tumor stromal fibroblasts was found to be associated with better overall survival and progression free survival [106]. The active VD metabolite calcitriol also inhibited pro-tumoral activation of NFs into CAFs, demonstrating the protective role of VD in CRC [106]. Finally, VDR was also found to inhibit activation of hepatic stellate cells into pro-tumor CAFs in hepatocellular carcinoma [107].

Based on the findings in pancreatic and colorectal cancers that demonstrate the ability of VD to inactivate CAFs and return them to a more quiescent phenotype [103,106], a recent study trialed a new approach to target CAFs in BC using FAP-coated nanoparticles loaded with the VD analog, calcipotriol (FAP-C NPs) [108]. In vitro, both free calcipotriol and FAP-C NPs induced the transition of BC CAFs into a quiescent state, as indicated by an increase in lipid droplets and a decrease in proteins associated with CAF activation (i.e., α-SMA, TGF-β and IL-6). When examined in vivo, FAP-C NPs inhibited tumor growth by 78.3% which was accompanied by a significant reduction in CAF-activation makers α-SMA and FAP and collagen levels in the tumor tissue. Remarkably, this approach also led to increased tumor infiltration of CD4+ and CD8+ cytotoxic T cells and increased pro-inflammatory cytokines IFN-γ and TNF-α in the serum. This is the first study to demonstrate that specific targeting of CAFs with VD is a viable approach for the treatment of BC and may inhibit CAF activation and improve the anti-tumor immune response [108].

The influence of VD on CAFs and their complex regulatory effects on gene expression within the TME is shown in Figure 3. Overall, the studies described above clearly demonstrate that VD exerts a dual effect on CAFs in BC and other cancers. However, further studies are required to unravel the complexity and molecular mechanisms behind these effects and how this relates to the CAF heterogeneity observed in BC.

### 5.2. Adipocytes

Adipocytes have been shown to play a significant role in the TME, being one of the main cellular constituent of the BC microenvironment [109]. Adipocytes can be activated by cancer cells into cancer associated adipocytes (CAAs) [110], which in turn actively support the development and spread of BC by secreting hormones, adipokines (i.e., adiponectin, leptin) and fatty acids [109]. The alteration of adipocyte phenotype by tumor cells exerts a strong effect on both cancer cells and the other components of the TME [111]. For instance, CAAs can actively recruit immune cells and polarize macrophages into the immunosuppressive M2 phenotype by secreting cytokines such as interleukin-6 (IL-6), chemokine ligand 2 (CCL2) and chemokine ligand 5 (CCL5). These effects promote BC malignancy by enhancing BC proliferation, angiogenesis, invasion, dissemination and metastasis [112].

An inverse relationship between adiposity and serum VD levels has been observed, suggesting that VD may influence health outcomes in obese BC patients [113]. Both breast and adipose tissues express VD receptors and the 1α-hydroxylase enzyme, enabling local conversion of VD into its active form. VD has been shown to modulate key aspects of adipocyte biology, including adipogenesis, differentiation, and inflammation [114]. Notably, Yum et al. demonstrated that conditioned media from VD-treated adipocytes significantly reduced the migration of MDA-MB-231 BC cells, indicating that VD alters the adipocyte secretome [115]. Specifically, VD suppressed the release of adipokines leptin and adiponectin, with the leptin:adiponectin ratio being significantly reduced, suggesting a lower degree of adipose tissue dysfunction. Treatment with VD also reduced the secretion of IGF-1 and proinflammatory cytokines (e.g., IL-6, CCL2, CX3CL1) from adipocytes, all of which contribute to BC cell migration [115].

VD supplementation has also been associated with reduced adiponectin expression, accompanied by decreased inflammation and tumor progression in mouse models of BC [116]. Furthermore, VD lowered the expression of inflammatory markers and limited macrophage infiltration into adipose tissue in mice fed a high-fat diet [117]. A potential mechanism by which VD can attenuate tumor progression may involve alterations in triacylglycerol hydrolysis, leading to increased glycerol release. Since glycerol has been implicated in promoting cancer progression in murine models [118], this pathway may be significant. Yum et al. also reported that VD treatment enhanced glycerol secretion by mature adipocytes, suggesting increased lipolysis [115]. This effect is likely mediated by enhanced phosphorylation of hormone-sensitive lipase, without concurrent fatty acid release [119].

The VD-mediated effects on adipocytes are likely facilitated through its binding to the VDR, directly influencing adipogenesis, metabolism, and inflammatory gene expression [120]. Through modulation of key signaling pathways, including downregulation of C/EBPα, C/EBPβ, PPARγ, and RXR, as well as activation of the WNT/β-catenin pathway, VD was shown to inhibit adipocyte differentiation [120]. Collectively, these findings highlight VD’s multifaceted role in modulating adipocyte–tumor cell interactions. By regulating adipokine secretion, reducing inflammation, and altering lipid metabolism, VD may suppress BC cell migration and progression through synergistic mechanisms.

### 5.3. Cancer Stem Cells

Accumulation of genetic and epigenetic alterations in tissue stem cells convert them into cancer stem cells (CSCs) [60,121], which are particularly involved in tumor initiation, progression, metastasis, recurrence and resistance to therapies [14]. However, CSCs constitute only a small fraction of cancer cells [14]. BC stem cells interact with the TME either directly or through the secretion of cytokines in a paracrine manner. Various cytokines/chemokines secreted by different components of the TME support the self-renewal of stem cells and aid them in escaping attack by the immune system [122]. BC stem cells exhibit enhanced upregulation of cell surface pumps to minimize drug accumulation, thereby resulting in a resistant tumor phenotype [123]. A recent study demonstrated that VD also has a role in regulation of cancer stemness by inhibiting proliferation of stem cells, suppressing stemness pathways and initiating differentiation of CSCs into less aggressive cell types (Figure 4) [60].

The transmembrane glycoprotein CD44 is a marker of BC stem cells and was correlated with drug resistance and increased BC cell migration and invasion [124]. Inducing the expression of CD44 in BC cells led to enhanced self-renewal and mammosphere growth [125]. Further, CD44+ BC cells were found to be enriched in residual BC following conventional therapies [126]. The BC cell line MCF10DCIS.com, which forms ductal carcinoma in situ (DCIS) lesions in vivo, expresses high levels of CD44, have tumor initiating properties and are often used by researchers as a model for BC stem cells [127]. Using this model, one study demonstrated that a class of synthetic VD analogs known as Gemini VD analogs, which have been structurally modified to enhance their biological effects, had potent anti-proliferative effects on the MCF10DCIS.com cells in vitro and in vivo [128]. This was accompanied by a significant reduction in CD44 expression in MCF10DCIS.com xenograft tumors, an effect that was found to be mediated by the VDR. In fact, the CD44 promoter region was found to possess a putative VDR element, suggesting that the VD-bound VDR may directly bind to the CD44 promoter to repress its expression [128].

Beyond its direct effects on CD44, VD has been shown to inhibit several pathways associated with cancer stemness. One key pathway is Wnt/β-catenin signaling, which plays a central role in CSC maintenance and function [129]. MCF-7 cells expressing the stem cell marker CD133 (CD133^+^ MCF-7) displayed greater resistance to tamoxifen and lower VDR expression compared to CD133^−^ counterparts, while also exhibiting elevated Wnt/β-catenin activity [130]. Notably, overexpression of VDR in CD133^+^ cells suppressed Wnt/β-catenin signaling and led to the formation of smaller, less viable spheroids [130], suggesting that VD may reduce cancer stemness and enhance tamoxifen sensitivity. Similar effects have been observed in ovarian CSCs, where VD inhibited Wnt signaling and reduced xenograft tumor growth [131]. In the MMTV-WNT1 transgenic mouse model of BC, Jeong et al. reported that VD not only suppressed tumor initiation and growth but also inhibited Wnt/β-catenin signaling and CSC generation [132]. Supporting this, in vitro studies demonstrated that VD impaired tumor-initiating cell spheroid formation in 3D culture assays [132].

Another crucial player in the maintenance and differentiation of stem cells is the Notch signaling pathway [129]. Its aberrant activation has been associated with development of different types of cancers including BC [133]. The Notch pathway consists of mammalian transmembrane receptors (Notch 1-4) and their membrane bounded ligands (JAG1, JAG2, δ-like ligand 1, 3 and 4) [134]. Upon binding to their ligands, Notch receptors undergo cleavage, release their intracellular domain, which leads to nuclear translocation and activation of target genes, including Hes1 [135], Hey1 [131], p21^CIP1^[135]^g^, cyclin D1 [132], c-myc [136] and NF-ĸB [137]. In normal mammary epithelial cells, constitutively active Notch receptor expression resulted in Notch signaling pathway activation. This led to dose-dependent hyper-proliferative responses and the formation of breast tumors [138]. Harrison et al. identified Notch-4 as target for reduction in BC recurrence [139]. Further, So et al. reported that VD’s suppressive effect on BC stem cells was dependent on Notch signaling inhibition mediated by Hes1 [128].

Recently, a small sub-population of human mammary epithelial cells (HMECs) when grown in suspension in vitro, was shown to be capable of forming three-dimensional (3D) spheroids called mammospheres [140]. Mammospheres are mainly composed of primitive mammary stem cells and progenitor cells with different proliferation potential and are capable of undergoing both self-renewal and multilineage differentiation in vitro [140]. Furthermore, when transplanted into mammary fat pads, these mammospheres were capable of generating whole mammary architecture in vitro, additionally verifying the existence of persistent self-renewing stem/progenitor populations within mammospheres [141]. Notably, VD was found to downregulate stem cell markers and genes related to the Notch signaling pathway to reduce the formation of mammospheres in BC [140]. In another study it was reported that an analog of VD, namely EB1089, inhibited mammosphere formation by BC cells [142]. This was further confirmed using VD analog BXL1024, which reduced the expression of genes involved in stem cell renewal, namely KLF4 and OCT4 in mammospheres [143]. However, two studies also showed that the expression of VDR was downregulated in BC mammospheres, suggesting that the effects of VD on BC stem cells may not be as potent as on other cells in the TME (i.e., CAFs, adipocytes and immune cells) [142,144].

### 5.4. Immune Cells

As described above, VD can influence inflammation and the immune response in BC via its effects on CAFs and adipocytes. However, VD can also exert direct effects on immune cells, playing a significant role in modulating the innate immune system, influencing the growth, differentiation and apoptosis of monocytes, dendritic cells, macrophages, natural killer cells, B cells, T cells as well as tumor infiltrating lymphocytes (TILs) in tumors and inflammation [14,145,146]. Immune cells express VDR and are capable of metabolizing VD. Notably, VDR expression in lymphocytes is induced once they are activated, while DCs and macrophages constitutively express the VDR [147]. The role of VD in modulating the innate and adaptive immune responses within the cancer microenvironment is further described below and summarized in Figure 5.

#### 5.4.1. Lymphocytes

Lymphocytes serve as key players in the adaptive immune system, orchestrating targeted responses against specific pathogens through immunological memory and antigen recognition. A recent study examining mice bearing EO771 BC orthotopic xenografts revealed that VD supplementation during tumor progression significantly reduced tumor growth and promoted cytotoxic CD8^+^ T cell infiltration into the tumors [116]. Interestingly, this effect was completely reversed in mice that were fed a high-fat diet, with VD supplementation inducing faster tumor progression and reducing infiltration of CD8^+^ T cells [116]. This suggests that VD metabolism in adipocytes plays an important role in determining its function. In fact, overweight mice were found to have increased levels of Cyp27a1, an enzyme that hydroxylates VD leading to higher local production of calcidiol in adipose tissue. It was hypothesized that the increase in adipocytes in overweight mice reduced calcidiol levels in the plasma, which may have then diluted its effect on the CD8^+^ T cells [116].

Apart from its effect on promoting CD8^+^ T cell tumor infiltration, VD was also found to inhibit the proliferation of T helper 17 (Th17) cells. Th17 cells promote BC growth and metastasis by producing the IL-17 cytokine [148]. VD was found to inhibit both human and murine T cells from producing IL-17, while enhancing their production of IL-4, which counteracts the pro-inflammatory effects of IL-17 [149,150]. The immunoregulatory function of VD was further demonstrated in vitro, using co-cultures of Th17 cells with MCF-7 cells. While Th17 stimulated the proliferation, migration and invasion of MCF-7 cells, VD introduction to this co-culture reversed the pro-tumor effects of Th17 cells (134). VD has also been shown to influence NK cell mediated cytotoxicity in BC. Min et al. showed that VD down-regulated the expression of miR-302c and miR-520c in MDA-MB-231 and MCF-7 cells, which in turn increased their susceptibility to NK cells [151].

#### 5.4.2. Myeloid Cells

VD was also found to influence immune cells of the myeloid lineage, including monocytes and macrophages, in the BC TME. These effects are highly context-dependent and appear to differ between human and murine models. In human BC patients, calcitriol has been reported to exert immunomodulatory effects by reducing pro-inflammatory activity and enhancing anti-tumor immune responses [152]. In contrast, studies in murine models have shown that VD can increase the production of pro-inflammatory cytokines such as IL-6 and IL-23, while simultaneously driving M2 polarization (likely due to COX-2/PGE_2_) of TAMs. Together, these effects suggest that VD may foster an immunosuppressive TME, thereby promoting tumor progression and metastasis [153].

Building on this, Anisiewicz et al. recently demonstrated that calcitriol and its analogs can influence tumor progression in either a pro-metastatic or anti-metastatic manner, depending on the age of the host organism [154]. In young mice, treatment with calcitriol and its analogs increased immunosuppression within the TME, whereas in aged, ovariectomized mice, the effect was reversed. These contrasting outcomes are likely due to age-related differences in immune system function [154], although they could also be linked to the loss of estrogen-VD synergy in the ovariectomized mice. In young mice bearing 4T1 mammary tumors, calcitriol elevated levels of the chemokine CCL2 in both plasma and tumor tissues [154]. CCL2 is known for its potent chemotactic activity, particularly in recruiting classical monocytes to sites of inflammation. It also promotes immune polarization toward a Th2 response and supports the differentiation of macrophages into the M2 phenotype, which is associated with tumor-promoting functions [155].

In mice, monocytes can be broadly categorized into two subpopulations: classical Ly6C^high^CX3CR1^low^CCR2^+^ monocytes (inflammatory) and non-classical Ly6C^low^CX3CR1^high^ monocytes (anti-inflammatory with tissue-repair functions) [156]. In the 4T1 mammary tumor model, treatment of mice with calcitriol and its analogs influenced the ratio of inflammatory vs. anti-inflammatory monocytes in a manner that was dependent on the age of the mice. For instance, in young mice treated with calcitriol, there were higher levels of anti-inflammatory Ly6C^low^ monocytes in the spleen and increased number of metastases in the lungs. However, in aged ovariectomized mice which represent the post-menopausal patient, there were fewer Ly6C^low^ monocytes in the spleen and fewer lung metastasis following treatment with calcitriol [154]. This suggests that VD may either promote or attenuate immunosuppression and metastasis in BC, and that these effects are dependent on other factors such as the efficiency of the immune system and hormonal context.

Another study examining murine bone marrow-derived macrophages indicates that VD may also exert immunosuppressive effects [157]. In this study, murine bone-marrow derived macrophages were differentiated into M0, M1, and M2 subtypes, with or without calcitriol treatment, and exposed to conditioned media (CM) from various murine mammary gland cell lines: 67NR (non-metastatic), 4T1 (metastatic), and Eph4-Ev (normal). Calcitriol treatment increased the expression of M2-associated markers such as *Spp1*, *Cd206*, CCL2, CD36, and arginase, while simultaneously decreasing M1-associated markers including *Cd80*, OPN, IL-1, IL-6, and iNOS [157]. The most pronounced effects were observed in M2 macrophages exposed to 4T1 CM, which significantly enhanced M2 polarization. Furthermore, M2 macrophages differentiated in the presence of calcitriol promoted the migration of both 4T1 and 67NR cells in vitro, suggesting a potential pro-metastatic influence of VD in the murine context [157].

To assess the effect of VD on the differentiation of human macrophages, a recent study isolated monocytes from BC patients and found that differentiating these in the presence of calcitriol led to reduced M1 and M2 markers [152]. Further, this study demonstrated that the expression of CYP27B1 in the tumor tissues of BC patients was negatively associated with CD200R, CD204 and CD44. This suggests that increased local capacity to activate VD via CYP27B1 leads to reduced TAM marker expression in human BC tumors, potentially contributing to VD’s anticancer activity [152].

The multifaceted nature of VD influence on immune components of the TME underscores its intricate involvement in immune system regulation. While evidence supports its dual role in immune modulation and immune suppression in cancer, further research using more physiologically relevant human models is warranted to elucidate the precise mechanisms governing these contrasting effects.

### 5.5. Extracellular Matrix

The extracellular matrix (ECM) is considered a significant regulator of BC carcinogenesis, with proteins constituting the ECM playing a significant role in cancer progression, metastasis and chemoresistance [121]. Intercommunication between tumor and ECM leads to activation of key signaling pathways which stimulates proliferation, invasion and metastasis [6]. The ECM comprises approximately 300 proteins including structural proteins such as collagen and elastin, adhesive proteins such as fibronectin and laminin, and proteoglycans such as hyaluronic acid [158]. Growth and survival of normal epithelial cells are dependent upon ECM attachment. Cellular changes like energy metabolism reprogramming in the primary tumors allows anchorage independence, which permits the survival of cancer cells detached from ECM, leading to growth of tumors and metastasis [159].

Matrix metalloproteinases (MMPs) are a family of zinc-dependent proteases that are considered to be crucial for the invasion, tumor angiogenesis and metastasis of BC, with MMP-2 and MMP-9 being correlated with worse patient prognosis [160]. The activity of MMPs can be inhibited by tissue inhibitors of metalloproteinases (TIMPs), which is necessary for the turnover of ECM for tissue remodeling in both pathological and physiological conditions [161]. Several studies have shown that VD has a role in regulating these MMPs and TIMPs [162]. For instance, prostate and BC cells treated with calcitriol had decreased MMP-9 and increased TIMP-1 expression, leading to inhibition of invasion [162]. This was further demonstrated therapeutically, with a micellar nano system containing calcitriol reducing BC tumor growth and metastasis via the down-regulation of MMP-2 and MMP-9, while at the same time up-regulating the focal adhesion protein paxillin [59].

Khuloud et al. also reported that BC cells, upon treatment with calcitriol, demonstrated increased expression of TIMP1 and TIMP2 while exhibiting reduced levels of MMP2 and MMP9 and their gelatinolytic activity [55]. Furthermore, calcitriol treatment led to diminished levels of amphiregulin, vascular endothelial growth factor (VEGF), and (TGF-β1), all of which promote vasculogenic mimicry (VM) formation [55]. VM are microvascular channels, which operate independently of endothelial cells and play a significant role in tumor progression by supplying blood to the tumor and contributing to resistance against chemotherapy [55]. High levels of MMPs are required for VM formation, as MMPs cleave laminin 5γ2 into 5γ2’ and 5γ2x fragments, which results in de novo vasculogenesis in solid tumors [163]. As a result of its inhibitory effects on MMP2 and MMP9, calcitriol was found to impede the formation of these VM channels in BC [55].

By reducing MMP-9 secretion, VD was also found to restore the epithelial state of cancer associated mesothelial cells (CAMs) by normalizing thrombospondin-1 (THBS1), thereby reducing ECM remodeling [164]. Mesothelial cells (MC) can undergo mesenchymal transition, transforming into CAMs and contributing to a tumor-supportive microenvironment that promotes metastasis [164]. Kitami et al. found that VD blocks the mesenchymal transition of MCs and suppresses THBS1 expression, an essential ECM molecule that facilitates cancer cell adhesion, ultimately reducing peritoneal dissemination [164]. Additionally, THBS1 also drives tumor invasion by stimulating the expression of MMPs through integrin signaling [165]. Thus, VD’s effect on the ECM plays a pivotal role in limiting CAM-driven metastatic potential.

Another ECM protein that was found to play an important role in BC is Tenascin C, which promotes invasion and growth of BC cells, while also stimulating angiogenesis in the TME during carcinogenesis [166]. Sancho et al. demonstrated that VD is capable of inhibiting tenascin C expression in human and mouse mammary epithelial cell lines with normal or malignant phenotypes [166]. Hyaluronan (HA) and hyaluronan synthesizing enzyme (HAS2) are also major components of the BC ECM, with higher HA/HAS2 being associated with aggressive disease and poor survival [12]. Carmen et al. demonstrated that HAS2 is a VD repressed gene, showing that VD not only inhibited the expression of HAS2 and HA synthesis, but also caused downregulation of BC stem cell marker CD44, which acts as HA receptor in Hs578T cells [167]. Conversely fibronectin, a non-collagenous adhesive glycoprotein that has a role in maintaining ECM and cell adhesion, was found to be negatively correlated with the metastatic potential of BC cells [168]. Polly et al. have shown that VD is able to regulate the transcription of fibronectin, leading to an increased expression of fibronectin through a DR6-type vitamin D-responsive element (VDRE) in its promoter [168]. Overall, these studies demonstrate the important role VD has in modulating ECM dynamics within the BC TME. The role of VD in modulating the ECM components within the tumor microenvironment is illustrated in Figure 6.

Considering the significant influence VD has on both BC cells and stromal components, it could serve as a promising supplement to current BC therapies to re-shape the TME and facilitate improved treatment efficacy. Indeed, several studies have already explored the potential use of VD in the treatment of BC, as described further in the section below. A summary of key in vitro studies evaluating the effects of VD on different BC cell lines is presented in Table 2.

## 6. Therapeutic Role of VD in the Prevention and Treatment of BC

Considering the well-established link between VD deficiency and the risk of developing BC, multiple clinical trials have been conducted to assess the effect of VD supplementation on BC progression and response to current treatments. Several studies report promising outcomes, including improved nutritional status, reduced inflammatory markers, and enhanced pathological response during chemotherapy (e.g., NCT03986268, NCT05331807). However, the overall efficacy of VD supplementation in BC has yielded mixed results, with some studies demonstrating that VD modulates inflammation in BC [102,116,117,149,154], an effect that is likely linked to its effects on different immune cell subsets within the BC TME, as described above in Section 4.4. A recent meta-analysis linked pre-treatment VD deficiency to poorer response to neoadjuvant chemotherapy, suggesting supplementation may improve prognosis in deficient patients [169]. Additional evidence indicates benefit for high-risk groups such as women with a BRCA mutation, who had 46% lower odds of BC when taking VD supplements [170]. Further, long-term calcium and VD supplementation reduced BC mortality in postmenopausal women [171].

Conversely, some studies report that although low VD levels correlate with higher BC risk, supplementation does not consistently prevent cancer, though it may improve outcomes in diagnosed patients [172]. These ambiguous results are likely impacted by multiple parameters including: (i)the dosing regimens and VD metabolites used, which differ significantly between studies [99,100,108,128,142,143,154]; (ii) the patients menopausal status, as VD was found to have differential effects in pre- vs. post-menopausal BC patients [101,173]; (iii) the BC subtype [174,175,176,177]; and (iv) whether there is an underlying VD deficiency, with patients that already have VD deficiency showing different responses to VD supplementation when compared to those that already have sufficient VD levels [101,102]. These results underscore the importance of understanding the molecular mechanisms by which VD influences both cancer cells and the TME in BC, as this knowledge could hold the key to targeting VD supplementation only to those patients that are likely to benefit from it.

From a clinical standpoint, the most pronounced effects are observed in patients who are VD deficient prior to treatment. VD supplementation in these patients has been linked to improved pathological responses, reduced inflammatory burden and an overall better prognosis. These observations highlight the need to assess baseline VD levels before supplementation to identify patients most likely to benefit from its therapeutic use.

Overall, our review of the literature suggests that VD supplementation offers the greatest benefit in patients with confirmed deficiency, a condition affecting up to 95% of newly diagnosed BC cases. Deficiency is associated with poorer prognosis, reduced chemotherapy response, and increased metastatic risk. Supplementation may be especially advantageous for high-risk groups, including those with aggressive subtypes such as triple-negative BC, BRCA mutation carriers, and postmenopausal women. In contrast, benefits in VD-sufficient patients remain inconsistent, reinforcing the importance of individualized approaches based on serum VD levels.

Other studies have assessed how VD can be used to reduce cardiotoxicity and musculoskeletal symptoms in BC patients, demonstrating a beneficial effect in managing side effects such as fatigue and joint pain (i.e., NCT00867217). Finally, multiple studies are currently in progress to assess the potential synergy when VD is combined with chemotherapy, although these results are yet to be published. A summary of all completed and ongoing clinical trials investigating the use of VD specifically in BC is shown in Table 3.

Recent studies have also highlighted the potential influence of the microbiome on efficacy of VD supplementation. For instance, Tirgar et al. investigated the effects of VD supplementation, alone and in combination with synbiotics (consisting of both pre- and pro-biotics) in BC patients. Their findings indicated that VD supplementation moderated the reduction in TNF-α levels, increased serum IL-6 levels, and, when combined with synbiotics, helped maintain IL-10 levels [178]. These results suggest a potential synergistic anti-inflammatory effect that could be beneficial in the supportive care of BC patients. Another study by Zhang et al. demonstrated that intestinal epithelial VDR has a crucial role in maintaining gut homeostasis and protects against BC by regulating microbial composition. While microbiota themselves does not express VDR, the authors showed that host intestinal VDR deficiency alters the gut microbial profile (dysbiosis), which in turn shifts breast-associated microbiota toward a carcinogenesis-promoting state. This dysbiosis, coupled with increased gut permeability and microbial translocation, exacerbates breast tumorigenesis [179]. Thus, the gut–tumor–microbiome axis, influenced by host intestinal VDR status, represents a potential therapeutic target for BC prevention and treatment, with the role of VD in this interaction remaining to be elucidated.

**Table 3 cancers-17-03751-t003:** A summary of clinical trials investigating the use of VD in BC. The table outlines an updated and curated overview of complete and ongoing clinical trials (ClinicalTrials.gov) assessing the role of VD in BC. This comprehensive overview provides insights into the diverse clinical approaches utilizing VD, highlighting its therapeutic potential across various study designs and outcomes.

NCT Number	Phase	Number of Patients	Parameters Studied	Intervention/Dosage	Menopause Status	Status	Results	Refs.
NCT04091178	II	57	This study evaluates the high dose efficacy of VD supplementation for BC treated adjuvant chemotherapy	25-OH VD (/one dose of 100,000 IU every 3 weeks)	Both premenopausal and postmenopausal women	Completed	High-dose VD supplementation found to correct deficiency, although some patients experienced asymptomatic grade 1 hypercalciuria.	[173,180,181,182,183,184,185,186,187,188,189,190,191]
NCT04166253	II	100	This study assesses the VD’s protective effect in doxorubicin induced toxicity in BC patients, Echocardiography changes and levels of biomarkers (VD, LDH, troponin-T and IL-6)	Alfacalcidol/ 0.5 mcg	Both premenopausal and postmenopausal women	Completed	No	No
NCT00656019	II	63	This study investigates whether levels of VD impact the characteristics of a woman’s breast cancer at diagnosis, and if low levels of VD change gene expression of their BC.	Calcipotriene/0, 2000, 4000, or 6000 IU per day orally	Both premenopausal and postmenopausal women	Completed	No observable effects	No
NCT01224678	III	300	This study assesses the effect of VD supplementation on mammographic density and to explore changes in the serum biomarkers (IGF1, atypia and Ki67)	VD/2000 IU	Premenopausal	Completed	−1.4% reduction in breast density	No
NCT03986268	NA	64	This study evaluates the relationship between the VD replacement and complete pathological response in patients undergoing neoadjuvant therapy	Cholecalciferol/50,000 IU weekly	Both premenopausal and postmenopausal women	Completed	VD supplementation during chemotherapy significantly improved pathological complete response rate, with a trend towards better overall survival.	[174]
NCT00976339	I	20	This study assesses the effect of high dose of VD on premenopausal women at high risk of developing BC	Cholecalciferol/20,000 IU, or 30,000 IU weekly)	Premenopausal	Completed	High dose of VD significantly increased the circulating levels of VD and have favorable effects on IGF signaling, but no notable changes in mammographic density.	[192]
NCT01948128	II	83	This study investigates the short-term effect of VD administration on BC clinical and translational markers	Cholecalciferol/40,000 IU	Both premenopausal and postmenopausal women	Completed	Higher VD levels have no significant effects on tumor proliferation or apoptosis	[193]
NCT01965522	II	100	Investigates the antiproliferative effect of VD and melatonin in BC (Ki67 expression and microRNA profile)	Melatonin/20 mg VD/2000 IU	Both premenopausal and postmenopausal women	Completed	No	[194]
NCT05331807	I	88	Evaluates the effect of co-supplementation of VD and omega-3 fatty acids on inflammatory biomarker	Omega-3 FA+ VD Supplementation/50,000 IU	Both premenopausal and postmenopausal women	Completed	Preliminary results show that combined omega-3 and VD supplementation can improve nutritional status and reduce inflammation markers.	[195]
NCT01747720	NA	405	Assesses the effect of VD supplementation in reducing mammographic breast density	Cholecalciferol/1000, 2000 or 3000 IU	Premenopausal	Completed	One year VD supplementation did not significantly reduce mammographic breast density compared to placebo in premenopausal women.	[196]
NCT01166763	NA	30	Explore the effect of VD dose (10,000 ul) in Mammographic Breast Density and expression of genes important in BC.	VD supplementation/10,000 IU	Premenopausal	Completed	Change in Mammographic Breast Density and a decrease as assessed by Ki-67 staining.	No
NCT01425476	I/II	45	Determine the effect of VD (cholecalciferol) alone and in combination with celecoxib on certain biomarkers of BC (PGE2, COX-2, and 15-PGDH)	Celecoxib/400 mg + Cholecalciferol/400 IU or 2000 IU	Both premenopausal and postmenopausal women	Completed	VD modulated molecular markers differently, resulting in distinct protective effects depending on BC risk level.	[197]
NCT01480869	III	215	This study aims to compare VD normalization after 6 months of baseline-adjusted versus standard supplementation	Calcium/500 mg and Cholecalciferol/100 000 IU	Both premenopausal and postmenopausal women	Completed	Tailored high-dose VD is a safe and more effective option for VD restoration in chemotherapy-treated BC patients, with better outcomes than conventional dosing.	[198]
NCT00859651	II	20	This phase II study enrolled 20 high-risk postmenopausal women to assess whether one year of high-dose VD (20,000 or 30,000 IU/week) increases serum levels and provides preliminary insights into its preventive effects against BC.	Cholecalciferol/20,000 IU or 30,000 IU	Postmenopausal women	Completed	Preliminary results show a significant increase in serum VD levels for both doses, and a favorable effect on IGF-1/IGFBP-3 ratio.	[192]
NCT00000611	III	68,135.	The study investigated the effects of hormone therapy, dietary changes, and calcium/VD supplementation on cardiovascular disease (CVD), BC, and osteoporosis in postmenopausal women.	VD supplementation/400 IU + calcium/1000 mg + hormone replacement therapy (Estrogen/progestin)	Postmenopausal women	Completed	The supplement regimen of calcium + vitamin D did not reduce overall mortality, but in long-term follow-up it appeared to reduce cancer deaths while increasing CVD deaths, with no net benefit on all-cause death.	[199]
NCT02274623	I	33	The study evaluated the safety and efficacy of CTAP101 in breast and prostate cancer patients with bone metastases, focusing on VD metabolism, serum calcium, bone pain, and quality of life.	CTAP101/30 μg + Calcifediol/1200 IU	Both premenopausal and postmenopausal women	Completed	No	No
NCT00867217		160	The study assessed whether high-dose VD3 reduces musculoskeletal symptoms in early-stage BC patients with low VD levels receiving letrozole.	VD supplementation/10,000 IU + Letrozole/2.5 mg	Both premenopausal and postmenopausal women	Completed	Adding high-dose VD3 to standard supplementation may help prevent worsening musculoskeletal symptoms and improve quality of life in early-stage BC patients on letrozole.	
NCT06551688	NA	184	The study investigates the association between preoperative VD levels and acute postoperative pain in BC surgery patients.	Serum VD levels/30 nmol/L	Both premenopausal and postmenopausal women	Completed	No	[30,33,185,200,201,202,203,204]
NCT01769625		31	The study evaluates the effects of VD alone and with celecoxib on BC risk biomarkers (PGE2, COX-2, 15-PGDH) and gene methylation in breast tissue.	Celecoxib/400 mg + cholecalciferol/2000 IU	Both premenopausal and postmenopausal women	Completed	VD decreases the PG cascade and increases TGFβ2 in a dose dependent manner, while adding celecoxib did not pro-vide synergy.	[197]
NCT01097278	IIB	208	The study evaluates the impact of high-dose cholecalciferol on mammographic density, breast tissue biomarkers, serum VD–related markers, and gene polymorphisms in premenopausal women at high risk for BC.	Cholecalciferol/400 IU daily	Both premenopausal and postmenopausal women	Completed	VD supplementation significantly increased the serum VD levels but had no significant effect on mammographic density	[205]
NCT03629717	I	10	The study examines the effects of RANKL inhibition with denosumab on breast tissue biomarkers in high-risk premenopausal women with dense breasts through daily VD (800 IU) and calcium (1200 mg) supplementation.	Denosumab/60 mg + Calcium/1200 mg +VD/800 IU	Premenopausal women	Completed	No	No
NCT01669343	NA	121	The study evaluates the impact of VD levels and BMI on estrogen suppression by letrozole in postmenopausal women, and assesses whether a higher letrozole dose enhances estrogen suppression in overweight/obese patients.	Letrozole/2.5 mg, 5 mg	Both premenopausal and postmenopausal women	Completed	Neither elevated BMI nor higher VD status impacted estrogen suppression by letrozole in the short term. Escalating the letrozole dose in women with BMI > 25 kg/m^2^ did not further suppress circulating estrogens during the 4-week intervention.	[206]
NCT01817231	NA	240	The study examined the association between serum 25(OH)D levels and BC risk in Saudi Arabian women through a case–control analysis.	Not provided	Both premenopausal and postmenopausal women	Completed	An inverse association exists between serum VD concentration and risk of BC	[207]
NCT04677816	II	50	The study investigates the effect of VD supplementation on pathological complete response in patients with VD deficiency and triple negative BC undergoing standard neoadjuvant chemotherapy	VD supplementation/50,000 IU	Both premenopausal and postmenopausal women	Recruiting	No	No
NCT06642441	NA	100	Evaluate the effect of VD supplementation on chemotherapy side effects following adjuvant chemotherapy in BC	Chemotherapy + VD2/10 mg	Both premenopausal and postmenopausal women	Recruiting	No	No
NCT04067726	II	210	The study determines the RANKL inhibition with denosumab, Calcium and VD to decrease mammographic density in high-risk premenopausal women with dense breasts.	Denosumab/60 mg + Calcium/500 mg + VD supplementation/400 IU	Both premenopausal and postmenopausal women	Active, not recruiting	No	No
NCT06596122	II	132	Exploring the neuroprotective effect of VD3 (Cholecalciferol) supplementation in conjunction with paclitaxel-based chemotherapy among BC patients with VD insufficiency or deficiency.	Cholecalciferol/100,000 IU	Both premenopausal and postmenopausal women	Not yet recruiting	No	No
NCT02856503	I/II	Not provided	Assessing the effect of high dose VD on the following biomarkers in the BC cells: VDR, estrogen receptor (ER), progesterone receptor (PR), epidermal growth factor receptor 2 (Her2/neu), androgen receptor (AR), as well as epidermal growth factor receptor 1 (EGFR) and Ki-67, as markers of proliferation, and E-cadherin, a marker of invasion and metastasis.	Toxiferol/ Cholecalciferol/50,000 IU	Both premenopausal and postmenopausal women	Withdrawn	No	No

### Future Prospects and Limitations of Current Evidence

While substantial preclinical and clinical evidence supports the role of VD in regulating the BC TME, several challenges remain before its full therapeutic integration. The effects of VD within the TME are highly context-dependent, showing both anti- and pro-inflammatory outcomes influenced by CAF heterogeneity, immune milieu, adiposity, menopausal status, and baseline VD levels. Moreover, discrepancies between murine and human studies add complexity to its clinical translation. Safety considerations are equally important, as prolonged high-dose VD supplementation carries risks such as hypercalcemia. Moving forward, deeper mechanistic insights into the cell-specific and context-dependent actions of VD are required, alongside the identification of predictive biomarkers for patient stratification. Combining VD with existing treatment modalities and exploring innovative delivery systems that maximize local efficacy while minimizing systemic toxicity represent promising directions. Collectively, these efforts may position VD as a safe and effective adjunct for reprogramming the BC TME and improving therapeutic outcomes.

Despite these compelling findings, the studied role of VD in modulating the TME of BC is limited by some experimental constraints. The use of supra-physiological concentrations of VD or its analogs may not accurately represent clinically acceptable doses. Moreover, the different responses of VD between murine and human models further complicates the translational interpretation. In vitro models like MCF-7, 4T1 and MMTV-PyMT cells, although informative, do not fully capture the complex heterogeneity of breast tumors. More studies focusing on use of physiologically relevant dosing, patient-derived models that incorporate key TME elements such as CAFs and immune cells, and integrative multi-omics studies are required to fill these gaps and validate the translational potential of VD use.

## 7. Conclusions

In this review, we provide a comprehensive description of the multifaceted role of VD in the TME of BC, encompassing its impact on immune cells, CAFs, CSCs, and the extracellular matrix. We further discuss how VD might influence the BC TME, and its potential utility as an anti-cancer agent which can help to re-shape the BC TME to enhance the therapeutic potential of current and emerging BC treatments. These insights underline VD’s implication in BC treatment paradigms and urge further exploration into its mechanisms for potential therapeutic interventions, emphasizing the need for future research aimed at unlocking the full potential of VD in combating BC progression and metastasis.

## Figures and Tables

**Figure 1 cancers-17-03751-f001:**
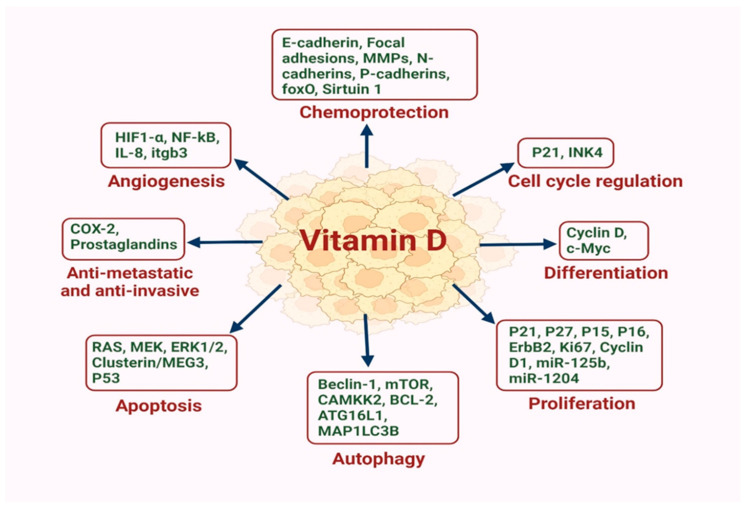
VD is emerging as a pivotal regulator in the intricate landscape of BC, exerting a multifaceted influence on multiple proteins and miRNAs that influence cell cycle regulation, differentiation, proliferation, autophagy, apoptosis, metastasis, angiogenesis and chemoprotection.

**Figure 2 cancers-17-03751-f002:**
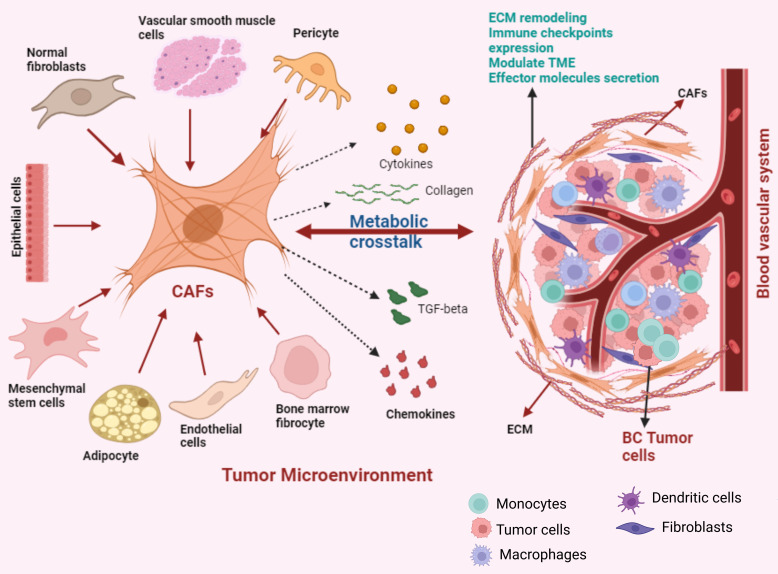
The dynamic transformation of diverse cell types into CAFs. These activated CAFs engage in metabolic crosstalk within the TME, influencing tumor progression and behavior.

**Figure 3 cancers-17-03751-f003:**
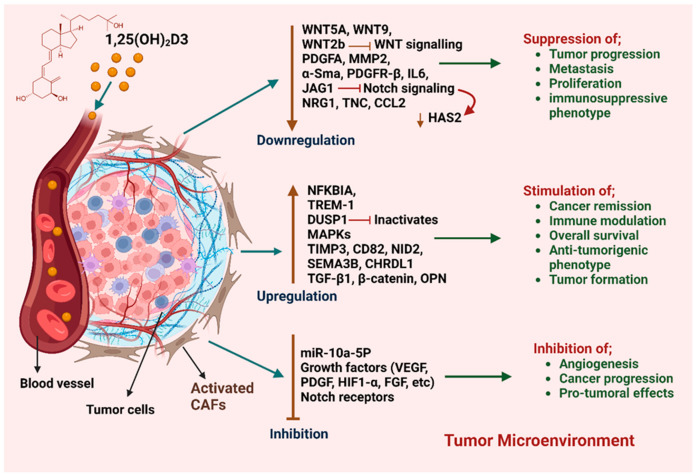
Impact of VD on CAFs and their regulatory effects on gene expression within BC TME. VD modulates multiple genes in CAFs, leading to diverse biological outcomes. While it plays a regulatory role, its effects remain complex, with some reports suggesting a potential tumor-promoting influence in certain contexts.

**Figure 4 cancers-17-03751-f004:**
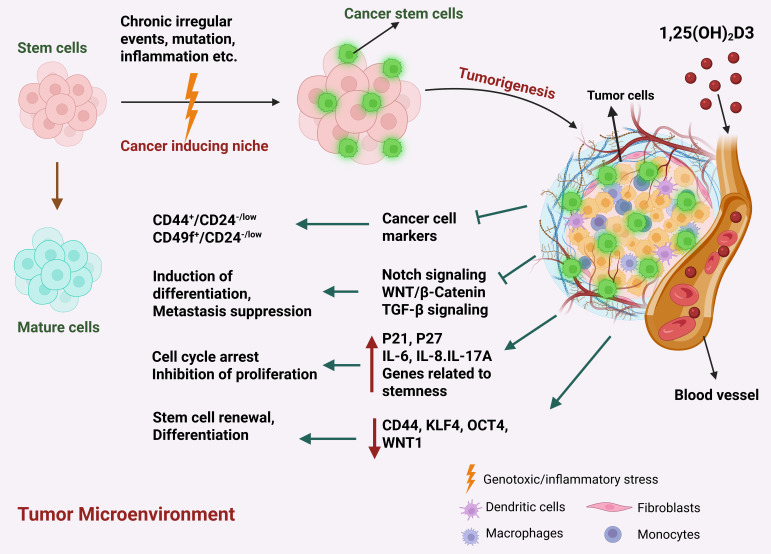
The intricate involvement of VD in the regulation of genes and markers associated with CSCs, influencing tumorigenesis and anti-cancer mechanisms within the TME. Highlighting the diverse actions of Vitamin D on CSC-related genes and markers, this illustration elucidates how VD intervention impacts CSC proliferation, differentiation, and self-renewal.

**Figure 5 cancers-17-03751-f005:**
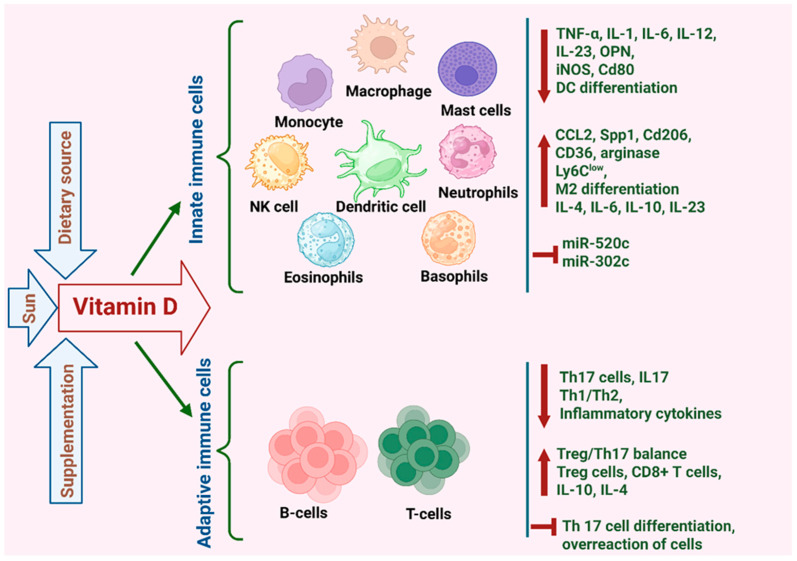
Highlighting VD’s pivotal role in modulating the innate and adaptive immune responses within the cancer microenvironment. This illustration outlines VD’s impact on diverse immune cells, including its regulation of downstream genes and cellular behavior.

**Figure 6 cancers-17-03751-f006:**
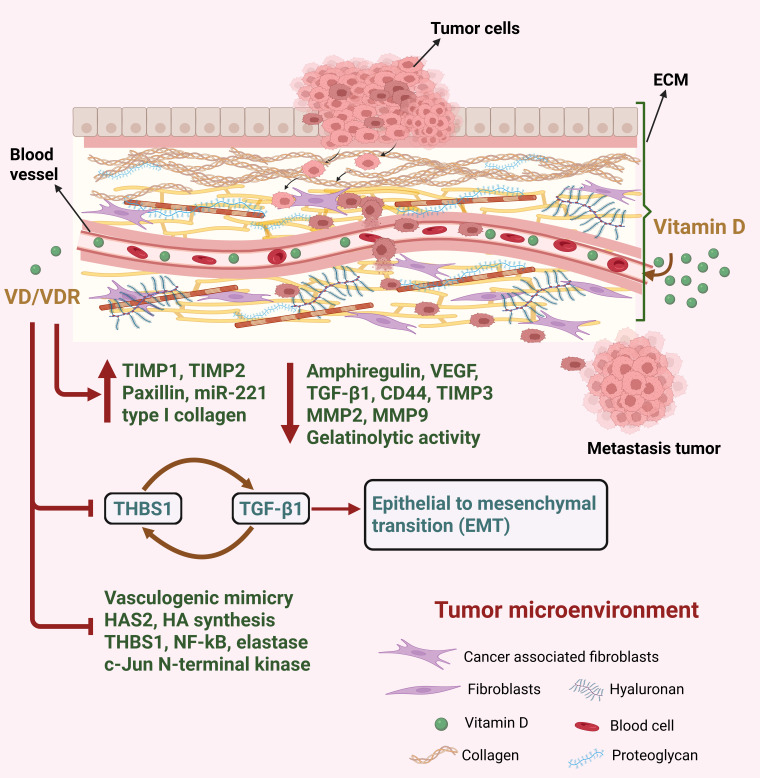
Illustrating VD’s regulatory role in modulating the ECM components within the TME. VD influences the expression of diverse proteins and genes involved in ECM remodeling, underscoring its potential in regulating tumor-stroma interactions, influencing metastasis, and modifying the TME’s structural and functional aspects.

**Table 1 cancers-17-03751-t001:** Summary of the anti-cancer effects of VD, outlining its involvement in various pathways and the regulation of specific proteins and genes.

Vitamin D Role	Pathways	Molecular Effects	Function	Refs.
Cell cycle	Cell cycle G_0_-S phase	Cyclin dependent kinases (CDKs)/ P21, INK4	Significantly alters cell cycle	[43]
Differentiation	WNT/β-catenin	E-cadherin/ cyclin D and c-Myc	Promotes cell differentiation	[65,66]
Proliferation	Cell death pathways	Ceramide kinase/ P15, P16, P21 and P27, ErbB2, Ki67, Cyclin D1, ceramide kinase, miR-125b, BAK-1, miR-1204	Anti-proliferative action	[48,49,50,67,68,69]
Apoptosis	Apoptotic pathways	Anti-apoptotic proteins/ BCL-2, RAS, MEK, ERK1/2, Clusterin/MEG3, p53	Apoptosis induction	[46,51,52,70]
Angiogenesis	NF-kB, EMT	Interleukin-8, HIF-1/ HIF-1 α, NF-kB, IL-8, itgb3	Inhibition of angiogenesis	[54,71,72]
Metastasis and invasion	Estrogen pathway	Aromatase enzyme/cyclooxygenase 2 COX-2, Prostaglandins, 15-hydroxy prostaglandins dehydrogenase	Prevents metastasis and invasion	[21,73]
Autophagy	ATG7 and Beclin-1 dependent pathway	Ca^2+^/calmodulin-dependent protein kinase kinase β and mTOR/ Beclin-1, AMPK, mTOR, MAP1LC38 (LC38), ATG16L1, BCL-2	Induction of autophagy	[61,62,63,64,74,75]

**Table 2 cancers-17-03751-t002:** Summary of in vitro studies evaluating the effects of VD on different BC cell lines demonstrating the mechanistic and phenotypic outcomes.

Cell Lines	Model Type	Effect of VD on Cell Lines	Subtype	References
MCF-7	Human	VD increased the sub-G0/G1 cell population and reduced S and G2/M phases, indicating dose-dependent growth inhibition.	ER+	[44,45]
MCF-7, MDA-MB-231	Human	VD significantly reduced the expression of the proliferation marker Ki-67.	ER+, TNBC	[46]
MCF-7, MDA-MB-231	Human	Inecalcitol combined with palbociclib significantly reduced tumor growth in ER+ breast cancer while having no effect on MDA-MB-231 in vivo.	ER+, TNBC	[47]
MCF-7, MDA-MB-231	Human	VD targets anti-apoptotic pathways, including inhibition of the RAS/MEK/ERK signaling pathway.	ER+, TNBC	[18]
MCF-7, MDA-MB-231	Human	VD suppresses metastasis by upregulating E-cadherin and downregulating mesenchymal markers P-cadherin and N-cadherin.	ER+, TNBC	[60]
MCF-10A, MCF-7, ZR-75-1, MDA-MB-453, MDA-MB-231	Human	VD activates VDR to regulate autophagy transcriptionally in breast cancer cells, promoting an autophagic signature linked to better prognosis.	ER+, ER+ PR+, ER-PR-AR+, TNBC	[61]
MCF-7	Human	VD induces autophagy and subsequent autophagy-dependent cell death in MCF-7 cells via increased cytosolic Ca^2+^ levels	ER+	[62]
MCF-7, MDA-MB-231	Human	VD may modulate CAF phenotype and their interaction with breast cancer cells	ER+, TNBC	[102]
MCF-7, BT-20, NPM-21T	Human	VD metabolite calcitriol inhibited cancer cell proliferation promoted by CAFs.	ER+, TNBC	[98]
MDA-MB-231	Human	VD-treated adipocytes reduced MDA-MB-231 cell migration, suggesting secretome modulation, reduced the secretion of IGF-1 and proinflammatory cytokines.	TNBC	[115]
MCF-7	Human	VD reduced cancer stemness and enhanced tamoxifen sensitivity by suppressing Wnt/β-catenin signalling.	ER+	[130]
MCF-7	Human	In Th17–MCF-7 co-cultures, vitamin D reversed the Th17-induced proliferation, migration, and invasion of MCF-7 cells	ER+	[133]
MCF-7 and MDA-MB-231	Human	VD enhances NK cell–mediated cytotoxicity in MCF-7 and MDA-MB-231 cells by downregulating miR-302c and miR-520c.	ER+, TNBC	[151]

## Data Availability

No new data were created.

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
