# Peer review of "The Effects of Vitamin D on the Breast Cancer Tumor Microenvironment"

_cancers, 2025, doi:10.3390/cancers17233751_

Round 1
Reviewer 1 Report
Comments and Suggestions for Authors
1. I would recommend adding a table that summarizes all cell lines and the effects of vitamin D.
2. It is difficult to evaluate the study results in Table 2 without adding information on the number of patients included in each study and the vitamin D dose, as the results will be highly dependent on the dosage.
3. Menopause status should also be clarified in Table 2.
4. The data on the effect of vitamin D supplements on breast cancer development were inconclusive, with only a minor benefit sometimes observed. So, in the author's opinion, what is the reason for this result?
Author Response
Comment 1: I would recommend adding a table that summarizes all cell lines and the effects of vitamin D.
Response: We appreciate the reviewer’s valuable suggestion. A summary table compiling all breast cancer cell lines and their observed responses to vitamin D treatment has now been added to the revised manuscript (now included as Table 2).
Comment 2: It is difficult to evaluate the study results in Table 2 without adding information on the number of patients included in each study and the vitamin D dose, as the results will be highly dependent on the dosage.
Response: We thank the reviewer for this insightful comment. The requested information regarding the number of patients enrolled in each study and the administered vitamin D dosage has been incorporated into Table 2 (now Table 3) to enhance clarity and interpretability of the study outcomes.
Comment 3: Menopause status should also be clarified in Table 2.
Response: We appreciate the reviewer’s suggestion. Information regarding the menopausal status of patients has been added to Table 2 (now Table 3) for improved clarity and completeness.
Comment 4: The data on the effect of vitamin D supplements on breast cancer development were inconclusive, with only a minor benefit sometimes observed. So, in the author's opinion, what is the reason for this result?
Response: We thank the reviewer for this insightful comment. The possible reasons for the inconclusive results regarding the effect of vitamin D supplementation on breast cancer development were discussed in Section 6:
“These ambiguous results are likely impacted by multiple parameters including: (i) the dosing regimens and VD metabolites used, which differ significantly between studies [99, 100, 108, 128, 142, 143, 154]; (ii) the patients menopausal status, as VD was found to have differential effects in pre- vs. post-menopausal BC patients [101, 173]; (iii) the BC subtype [174–177]; and (iv) whether there is an underlying VD deficiency, with patients that al-ready have VD deficiency showing different responses to VD supplementation when compared to those that already have sufficient VD levels [101, 102].”
We have now added further discussion on this important point from a more clinical perspective:
“From a clinical standpoint, the most pronounced effects are observed in patients who are VD deficient prior to treatment. VD supplementation in these patients has been linked to improved pathological responses, reduced inflammatory burden and an overall better prognosis. These observations highlight the need to assess baseline VD levels before supplementation to identify patients most likely to benefit from its therapeutic use.
Overall, our review of the literature suggests that VD supplementation offers the greatest benefit in patients with confirmed deficiency, a condition affecting up to 95% of newly diagnosed BC cases. Deficiency is associated with poorer prognosis, reduced chemotherapy response, and increased metastatic risk. Supplementation may be especially advantageous for high-risk groups, including those with aggressive subtypes such as triple-negative BC, BRCA mutation carriers, and postmenopausal women. In contrast, benefits in VD-sufficient patients remain inconsistent, reinforcing the importance of individualized approaches based on serum VD levels.”
Reviewer 2 Report
Comments and Suggestions for Authors
With a focus on stromal and immune components of the tumor microenvironment (TME), such as CAFs, adipocytes, CSCs, immune cells, and the extracellular matrix (ECM), the manuscript titled "The effects of Vitamin D on the breast cancer tumor microenvironment" offers a thorough narrative overview of how vitamin D (VD) influences the biology of breast cancer (BC).
The manuscript is generally informative and well-structured, and the topic is current and pertinent. However, before the work is ready for publication, a number of important issues need to be resolved.
-Although it is presented as a thorough review, the manuscript lacks a clear section outlining the methods used to find, choose, and synthesize the literature (databases searched, time frame, key words, inclusion/exclusion criteria, language restrictions, e.t.c.)
-Sharpening the clinical perspective would be beneficial by: Making educated guesses about which subgroups might benefit the most (e.g., deficient patients, specific subtypes, obese vs. lean, pre- vs. post-menopausal), highlighting the possible function of VDR expression in tumor/tumor stroma and/or serum 25(OH)D as stratification biomarkers. From a clinical standpoint, which patients stand to gain?
- Ex vivo CAF/adipocyte systems, high VD analog doses, or murine models are used in a number of important mechanistic sections. The following should be specifically acknowledged in order to improve the manuscript: The problem of supra-physiological calcitriol or analog concentrations, Immune/TME responses vary by species. certain cell line models' artificial nature (e.g. MCF-7, 4T1, MMTV-PyMT, etc.). This would be covered in a brief paragraph in the Conclusion or Future Prospects titled "Limitations of current evidence."
Author Response
Comment 1: Although it is presented as a thorough review, the manuscript lacks a clear section outlining the methods used to find, choose, and synthesize the literature (databases searched, time frame, key words, inclusion/exclusion criteria, language restrictions, e.t.c.)
Response: We appreciate the reviewer’s observation. A new Methodology section (Section 2) has now been added to the revised manuscript, detailing the databases searched, search strategy, keywords, inclusion and exclusion criteria, and other relevant parameters.
Comment 2: Sharpening the clinical perspective would be beneficial by: Making educated guesses about which subgroups might benefit the most (e.g., deficient patients, specific subtypes, obese vs. lean, pre- vs. post-menopausal), highlighting the possible function of VDR expression in tumor/tumor stroma and/or serum 25(OH)D as stratification biomarkers. From a clinical standpoint, which patients stand to gain?
Response: We thank the reviewer for this valuable suggestion. The clinical perspective has been expanded in the revised manuscript (Section 6) to address this point:
“From a clinical standpoint, the most pronounced effects are observed in patients who are VD deficient prior to treatment. VD supplementation in these patients has been linked to improved pathological responses, reduced inflammatory burden and an overall better prognosis. These observations highlight the need to assess baseline VD levels before supplementation to identify patients most likely to benefit from its therapeutic use.
Overall, our review of the literature suggests that VD supplementation offers the greatest benefit in patients with confirmed deficiency, a condition affecting up to 95% of newly diagnosed BC cases. Deficiency is associated with poorer prognosis, reduced chemotherapy response, and increased metastatic risk. Supplementation may be especially advantageous for high-risk groups, including those with aggressive subtypes such as triple-negative BC, BRCA mutation carriers, and postmenopausal women. In contrast, benefits in VD-sufficient patients remain inconsistent, reinforcing the importance of individualized approaches based on serum VD levels.”
Comment 3: Ex vivo CAF/adipocyte systems, high VD analog doses, or murine models are used in a number of important mechanistic sections. The following should be specifically acknowledged in order to improve the manuscript: The problem of supra-physiological calcitriol or analog concentrations, Immune/TME responses vary by species. certain cell line models' artificial nature (e.g. MCF-7, 4T1, MMTV-PyMT, etc.). This would be covered in a brief paragraph in the Conclusion or Future Prospects titled "Limitations of current evidence.
Response: We appreciate the reviewer’s insightful comment. A brief paragraph addressing the limitations of current evidence has now been added to section 6.1, which has been renamed “Future prospects and limitations of current evidence”
“Despite these compelling findings, the studied role of VD in modulating the TME of BC is limited by some experimental constraints. The use of supra-physiological concentrations of VD or its analogs may not accurately represent clinically acceptable doses. Moreover, the different responses of VD between murine and human models further com-plicates the translational interpretation. In vitro models like MCF-7, 4T1 and MMTV-PyMT cells, although informative, do not fully capture the complex heterogeneity of breast tumors. More studies focusing on use of physiologically relevant dosing, patient-derived models that incorporate key TME elements such as CAFs and immune cells, and integrative multi-omics studies are required to fill these gaps and validate the translational potential of VD use.”
Round 2
Reviewer 1 Report
Comments and Suggestions for Authors
I have no more comments on the manuscript.